# The Application of Structural Retinal Biomarkers to Evaluate the Effect of Intravitreal Ranibizumab and Dexamethasone Intravitreal Implant on Treatment of Diabetic Macular Edema

**DOI:** 10.3390/diagnostics10060413

**Published:** 2020-06-17

**Authors:** Ida Ceravolo, Giovanni William Oliverio, Angela Alibrandi, Ahsan Bhatti, Luigi Trombetta, Robert Rejdak, Mario Damiano Toro, Costantino John Trombetta

**Affiliations:** 1Institute of Ophthalmology, Department of Biomedical Sciences, University of Messina, 98124 Messina, Italy; g.w89@me.com (G.W.O.); aalibrandi@unime.it (A.A.); luigiunime@libero.it (L.T.); ctrombetta@unime.it (C.J.T.); 2Glangwili General Hospital, Carmarthen SA31 2AF, Wales, UK; ahsan.alam.bhatti@gmail.com; 3Department of General Ophthalmology and Pediatric Ophthalmology Service, Medical University of Lublin, 20079 Lublin, Poland; robertrejdak@yahoo.com; 4Faculty of Medical Sciences, Collegium Medicum Cardinal Stefan Wyszyński University, 01815 Warsaw, Poland

**Keywords:** optical coherence tomography, diabetic macular edema, biomarkers, inflammation

## Abstract

Background: The aim of this study was to compare the therapeutic effect of intravitreal treatment with ranibizumab and dexamethasone using specific swept-source optical coherence tomography retinal biomarkers in patients with diabetic macular edema (DME). Methods: 156 treatment-naïve patients with DME were divided in two groups: 75 patients received 3 monthly intravitreal injections of ranibizumab 0.5 mg (Lucentis^®^) (Group 1) and 81 patients received an intravitreal implant of dexamethasone 0.7 mg (Ozurdex^®^) (Group 2). Patients were evaluated at baseline (V1), at three months post-treatment in Group 1, and at two months post-treatment in Group 2 (V2). Best-corrected visual acuity (BCVA) and swept source-OCT were recorded at each interval. Changes between V1 and V2 were analyzed using the Wilcoxon test and differences between the two groups of treatment were assessed using the Mann–Whitney test. Multiple regression analysis was performed to evaluate the possible OCT biomarker (CRT, ICR, CT, SND, HRS) as predictive factors for final visual acuity improvement. Results: In both groups, BCVA improved (*p*-value < 0.0001), and a significant reduction in central retinal thickness, intra-retinal cysts, red dots, hyper-reflective spots (HRS), and serous detachment of neuro-epithelium (SDN) was observed. A superiority of dexamethasone over ranibizumab in reducing the SDN height (*p*-value = 0.03) and HRS (*p*-value = 0.01) was documented. Conclusions: Ranibizumab and dexamethasone are effective in the treatment of DME, as demonstrated by functional improvement and morphological biomarker change. DME associated with SDN and HRS represents a specific inflammatory pattern for which dexamethasone appears to be more effective.

## 1. Introduction

Diabetic retinopathy (DR) is the most frequent complication of diabetes and represents one of the leading causes of legal blindness, primarily as a result of diabetic macular edema (DME) [1,2]. DME occurs due to an abnormal intra-retinal and eventually sub-retinal fluid collection in the macular area caused by the alteration of the blood retinal barrier and is characterized by pericyte loss and endothelial cell junction breakdown [3].

In current practice, fluorescein angiography (FA) and optical coherence tomography (OCT) are the techniques of choice for evaluating diabetic maculopathy, providing several quantitative and qualitative data points concerning DME, which have been extensively used in randomized clinical trials and in clinical practice as non-invasive biomarkers [4].

In particular, central retinal thickness (CRT), the presence of hyper-reflective spot (HRS), serous detachment of neuro-epithelium (SDN), the size of intra-retinal cysts (IRC), the occurrence of disorganization of the inner retinal layers (DRIL), the state of the ellipsoid zone (EZ), the external limiting membrane (ELM), as well as the choroidal thickness (CT) have been used to characterize DME [4,5,6,7,8].

Recent animal and clinical studies strongly suggest a central inflammatory role in DME. Multiple cytokines and chemokine are involved in the pathogenesis of DME, with several cellular processes affecting the neurovascular unit [9,10]. Specifically, SDN and HRS have been recently proposed as non-invasive OCT-imaging biomarkers of retinal inflammation in DME [11].

Currently, the predominant treatments for DME are laser photocoagulation, intravitreal injections of corticosteroids, or anti-VEGF agents [12,13]. Numerous factors influence the choice of anti-VEGF or corticosteroids as first-line treatment in naïve patients with DME, including crystalline status, the intraocular pressure, recent cardiovascular event, and the age of the patients [7]. However, the presence of specific structural OCT biomarkers may help additionally guide the choice of treatment and monitor therapeutic response.

The aim of this study was to compare the short-term effects of intravitreal ranibizumab and an intravitreal dexamethasone implant in the treatment of DME by evaluating changes in specific structural OCT biomarkers.

## 2. Materials and Methods

In this retrospective study, we analyzed the data of 156 treatment-naïve patients (156 eyes) with center-involving DME.

Two groups of patients were established: Group 1 (75 eyes) were treated with three monthly intravitreal injections of ranibizumab 0.5 mg (Lucentis^®^, Novartis, Genentech, San Francisco, CA, USA) and Group 2 (81 eyes) were treated with an intravitreal implant of dexamethasone 0.7 mg (Ozurdex^®^, Allergan. Inc., Irvine, CA, USA).

The inclusion criteria were: age > 18 years, treatment-naïve DME with central macular thickness (CMT) on SS-OCT ≥ 300 µm, and best-corrected visual acuity (BCVA) between an Early Treatment Diabetic Retinopathy Study (ETDRS) score of 35 and a score of 80.

The exclusion criteria were: comorbidity with other retinal diseases, history of prior retinal surgery, the presence of vision-impairing cataract, other maculopathy, glaucoma or ocular hypertension, iris neovascularization, and any other ocular condition that can influence visual acuity. The choice of treatment was in accordance with the recent guidelines proposed by the European Society of Retina Specialists (EURETINA) [12]. As first-line therapy, we considered the use of corticosteroids in patients who had a history of a major cardiovascular event, poor compliance to monthly anti-VEGF treatment, and preferably pseudophakic patients or those without crystalline lens opacity. Anti-VEGF was preferentially considered in phakic patients in the absence of cardiovascular contraindications.

Informed consent was obtained from each patient, and the research was carried out in accordance with the Declaration of Helsinki, at the Retinal service of the Department of Biomedical Sciences, University of Messina, Italy, between January 2018 and April 2019. Approval from the Ethics Committee of the University of Messina was obtained for this study (protocol number 78/18, approved on: 19 November 2018).

Anamnestic information was documented for each patient, including type of DM, duration of DM, value of glycated hemoglobin (HbA1c), and type of DM therapy.

A complete bilateral eye examination, including best-corrected visual acuity (BCVA), slit lamp microscope examination, Goldmann applanation tonometry, fundus color photography, red-free fundus photography, and swept-source OCT (DRI SS-OCT Triton plus, Topcon Corporation, Tokyo, Japan) were performed.

This data was gathered at each visit, prior to the first injection (baseline), 1 month after completing the loading dose with 3 monthly injections in Group 1, and after 2 months from the treatment in Group 2. The main outcomes of this study were to evaluate the changes in OCT biomarkers (CRT, IRC, CT, SDR, HRS), in the early months after treatment, at the time of the maximum pharmacological effect of these drugs [13,14].

### 2.1. Retinal Images

All SS-OCT images were acquired using a 6-mm radial OCT scan (consisting of 12 scans 15° apart) centered on the fovea by an experienced technician (G.W.O). Automatic analysis by OCT software (IMAGEnet 6) was performed to evaluate the CRT. A manual count of red dots was assessed on red-free and color fundus photographs inside an ETDRS grid. The total number of red dots, the number of newly formed red dots, and the number of red dots which had disappeared was documented by comparing the baseline with the last follow-up visit (Figure 1A).

A manual count of the total number of hyper-reflective spots (HRS), defined as small (<30 mm), punctiform discrete white lesions with reflectivity similar to the nerve fiber layer, absence of back-shadowing, and location in both inner and outer retina, was performed between the two vertical lines and calculated in the area of 3000 µm centered on the fovea (Figure 1B) [15].

Intra retinal cysts (IRC) appear as round or elongated spaces with low-reflectivity inside the neuro-retina. The height of each individual cyst was measured in the area of 3000 µm, centered on the foveal center and summed to obtain the composite value of IRC (Figure 1C) [15].

The choroidal layer exists between the hyper-reflective line just below the RPE corresponding to Bruch’s membrane (BM) and the choroidal-scleral junction. Choroidal thickness (CT) was assessed as the mean value of the measurement in three points, at the foveal center, and at 1500 µm temporally and 1500 µm nasally, respectively from this (Figure 1D).

The height of serous detachment of neuro-epithelium (SDN) was manually measured using the built-in calliper function of the OCT device, as the distance between the outer retinal and the pigment epithelial surfaces at the foveal center (Figure 1E).

### 2.2. Statistical Analysis

The numerical data are expressed as mean and standard deviation and the categorical variables as absolute frequency and percentage. Examined variables did not present a normal distribution as verified by the Kolmogorov–Smirnov test and therefore, a non-parametric approach has been used. For each parameter, we performed statistical comparisons between groups using the Chi Square test for categorical variables and Mann–Whitney test for numerical variables. For each group, in order to evaluate the existence of statistically significant differences in different times of observation, we applied the Wilcoxon test (for numerical parameters) and the McNemar test (for dichotomous variables). Multiple regression analysis was performed to evaluate the possible OCT biomarker (CRT, ICR, CT, SND, HRS) as predictive factors for final visual acuity improvement at the end of treatment.

A linear regression analysis was performed for SDN height, BCVA, HRS. A *p*-value smaller than 0.05 was considered to be statistically significant. Statistical analyses were performed using the SPSS 22.0 for Windows package.

## 3. Results

A total of 156 patients (83 males, 73 females; mean age 59.0 ± 8.9 years) were included in the study. All patients were treatment-naïve, 75 received three monthly intravitreal injections of ranibizumab (Group 1), and 81 patients received a dexamethasone implant (Group 2). Table 1 summarizes the baseline characteristics and demographic data of the patients.

These groups were similar in terms of gender, age, HbA1c levels, type of diabetes, duration of diabetes, BCVA, and SS-OCT data evaluated at baseline.

Baseline and post-treatment data are reported in Table 2 and Table 3 and Figure 2 and Figure 3.

The mean BCVA improved from 51.6 ± 17.1 to 56.9 ± 17.3 ETDRS letters (*p* < 0.001) in Group 1, and from 47.8 ± 16.8 to 55.4 ± 16.8 ETDRS letters in Group 2 (*p* < 0.001) after treatment. Additionally, in Group 1, there were statistically significant changes in CRT, IRC, HRS, and CT. A significant reduction of red dots evaluated after treatment was noted (*p* < 0.001). Eighteen patients (24.0%) treated with ranibizumab showed an SDN at baseline. The mean height of SDN changed after treatment from 100.5 ± 59.5µm to 63.1 ± 46.8 µm, and this was statistically significant (*p* = 0.02). In Group 2, a significant change in CRT, IRC, HRS, CT, and the number of red dots observed after 2 months from the dexamethasone implant were recognized.

Eighteen patients (22.2%) in Group 2 presented an SDN at baseline SS-OCT; after treatment, a statistically significant decrease (from 108.7 ± 47.4 µm to 19.4 ± 23.5 µm) of SDN height was observed (*p* = 0.002).

Table 4 shows a significant change in SDN (*p* = 0.03) and HRS (*p* = 0.01) between the two treatment groups after treatment. At the end of treatment, 4 patients in Group 1 (22.2%) and 16 (88.8%) patients in Group 2 presented a complete resolution of SDN (*p* = 0.0002).

We used multiple regression analysis to evaluate possible OCT biomarkers predicting factors of improvement of BCVA at the end of treatment. In particular, a reduction of the CRT value at baseline was significantly associated with an improvement of BCVA > 10 letters ETDRS at the end of treatment (*p* = 0.0001), as well as SND height at baseline (*p* = 0.032) (Table 5).

A negative correlation between baseline SDN height and BCVA was observed (coefficient = −0.738 *r*^2^ = 0.544, *p* = 0.0001) and BCVA post-treatment (coefficient = −0.688, *r*^2^ = 0.473, *p* = 0.0001). Conversely, a positive correlation between baseline SDN height and baseline number of HRS was seen (coefficient = 0.645, *r*^2^ = 0.416, *p* = 0.0001) (Figure 4).

## 4. Discussion

Numerous studies have been conducted on patients with DME treated with anti-VEGF injections, and it is established that microstructural changes visible at baseline through OCT scans might predict the response to treatment [5,11,12,13,14,15,16]. However, there is a paucity of literature regarding such prognostic indicators after treatment with dexamethasone implants [17,18]. Moreover, it is established that several factors can influence treatment outcomes in DME, including duration and extent of edema, pattern of edema, and characteristics of a single retina layer structure/thickness [5,11,12,13,14,15,16].

This study compared two groups of treatment-naïve patients with DME, treated with a ranibizumab or dexamethasone implant, evaluating the changes in several morphological OCT biomarkers and functional parameters to assess treatment response.

In our findings, both groups showed a significant improvement in BCVA as well as a reduction of CRT, IRC, and red dots after treatment. However, patients treated with dexamethasone showed an additional major effect on reducing HRS number (*p* = 0.03), and height of SDN (*p* = 0.01) compared to ranibizumab. Underlying cellular processes, in particular the significance of inflammation in the formation of these microstructures, may explain our results.

Firstly, the pathophysiology of SDN is not completely understood. It is proposed to be due to the migration of fluid from the intra-retinal cystoid space, or directly from choroidal circulation to the sub-retinal space, thereby exceeding reabsorption capacity of the RPE. Other factors such as impairment of choriocapillaris flow and the integrity of the external limiting membrane (ELM) may play a role in the pathogenesis of SDN [19,20]. However, more recently, studies have also suggested that inflammation plays a central role in alteration of the neurovascular unit and control of fluid transport [9,10].

Other mediators of inflammation have also been implicated in the pathogenesis of DME, including multiple cytokines and chemokines that are linked to metabolic states, such as hyperglycemia. Hyperglycemia is the strongest risk factor contributing to the pathogenesis of DME, leading modifications in biochemical pathways including diacylglycerol (DAG)–protein kinase C (PKC), advanced glycation end products/receptor for advanced glycation end products, polyol (sorbitol), and hexosamine pathways [21]. All these metabolic pathways cause an increase in inflammation and oxidative stress, upregulation of cytokines and growth factors such as interleukins (ILs), angiopoietins, tumor necrosis factor, matrix metalloproteinases, and vascular endothelial growth factor (VEGF) [9,10].

Several studies have proposed SDN and HRS as non-invasive OCT-imaging biomarkers for the identification of retinal inflammation in DR and DME. Indeed, DME with SDN is considered a distinct morphologic pattern of DME associated with a higher concentration of inflammatory cytokines, specifically IL-6 in the vitreous, when compared with DME without SDN [22]. SDN occurs in up to 31% of patients with DME, and has been associated with decreased retinal sensitivity, increased choroidal thickness, and a disrupted external limiting membrane (ELM) [11,23]. By comparison, in our study, SDN occurred globally in 23% of the patients evaluated. Some studies have reported a poorer visual outcome after treatment of DME with SDN [24]. In the sub-group of DME with SDN in our study, there was an incremental increase in BCVA from 51.9 to 55.6 letters (*p* = 0.008), and from 47.5 to 51.2 letters (*p* = 0.035), in patients treated with dexamethasone and ranibizumab, respectively.

The differences in outcomes and SDN changes reported in our study between ranibizumab and dexamethasone may be related to the latter’s therapeutic mechanism. Corticosteroids induce anti-inflammatory effects by inhibiting retinal VEGF, ICAM-1, and TNF-α expression levels as well as increasing the expression of anti-inflammatory agents, such as IL-10 and adenosine [25]. In addition to their anti-inflammatory effects, corticosteroids induce retinal fluid clearance via the transcellular aquaporin-4 and potassium channels in retinal Muller cells [26].

In our study, a complete resolution of SDN was noted in 88.8% of patients treated with dexamethasone after 3 months, as well as a statistically significant reduction of SDN height (*p* = 0.0002).

In the literature, only a few studies with small sample sizes have compared dexamethasone with ranibizumab in the treatment of SDN-associated DME [11,17,26]. Vujosevic et al. reported no significant differences in SRD resolution between the two treatment groups but observed a trend toward a higher response in the dexamethasone group [15]. We documented a superiority of dexamethasone over ranibizumab in reducing the SRD height (*p*-value = 0.03).

Like SDN, HRS have been suggested to represent activated aggregates of microglial cells and are also considered an imaging biomarker of retinal inflammatory response [27,28]. Additionally, the number of HRS has been associated with different stages of DR and DME severity [29]. The mean number of HRS was 66.4 ± 18.6 in Group 1, and 68.1 ± 19.8 in Group 2. In each group, a reduction of HRS was observed after treatment, but it was significantly greater in the dexamethasone group (*p* = 0.01).

Since the advent of SS-OCT, it has been suggested that there may be a reduction of CT in DR compared with controls. Furthermore, in the foveal region, the choroid of diabetic patients appears to be thinner in eyes with signs of DR than in eyes without DR [30]. A CT reduction after treatment with intravitreal bevacizumab injection and pan-retinal photocoagulation has been previously shown, suggesting that this treatment rapidly decreases choroidal vascular permeability or causes atrophy of choroidal vessels [5,31,32]. In our study, a reduction of CT was observed in both treatment groups, from 261.1 ± 33.1 µm to 259.2 ± 33.3 µm (*p*-value = 0.11) in Group 1, and from 271.5 ± 64.7 µm to 242.9 ± 60.7 µm (*p*-value < 0.0001) in Group 2.

Recent studies have emphasized the turnover of red dots as a predictive factor for the progression to DME and to identify patients being at risk of developing sight-threatening complications [33,34]. In one study, the turnover of red dots in a group of patients who had already developed DME was shown to decrease after being treated with an anti-VEGF drug (ranibizumab). This led to the conclusion that the treatment of macular edema using ranibizumab has an impact on the turnover of red dots in diabetic retinopathy [35]. In our study, a significant reduction of red dots was observed not only in the ranibizumab group (*p*-value < 0.0001), but also in the dexamethasone group (*p*-value < 0.0001).

The retrospective design, small sample size, as well as the short follow-up periods represent the main limitations of this study.

## 5. Conclusions

Ranibizumab and dexamethasone are both effective in the treatment of DME, as demonstrated by functional improvement and morphological biomarker change. CRT represents the principal OCT structural biomarkers predicting the visual outcome at the end of treatment.

Moreover, as already documented in previous studies [11,15,22,36,37], we confirmed that DME associated with SDN and a high number of HRS describes a specific inflammatory pattern of DME and shows a better response to intravitreal steroids than to anti-VEGF treatment.

These results may be helpful in clinical practice, by enabling the recognition of specific non-invasive imaging biomarkers that can be used to support the measurement of treatment response.

## Figures and Tables

**Figure 1 diagnostics-10-00413-f001:**
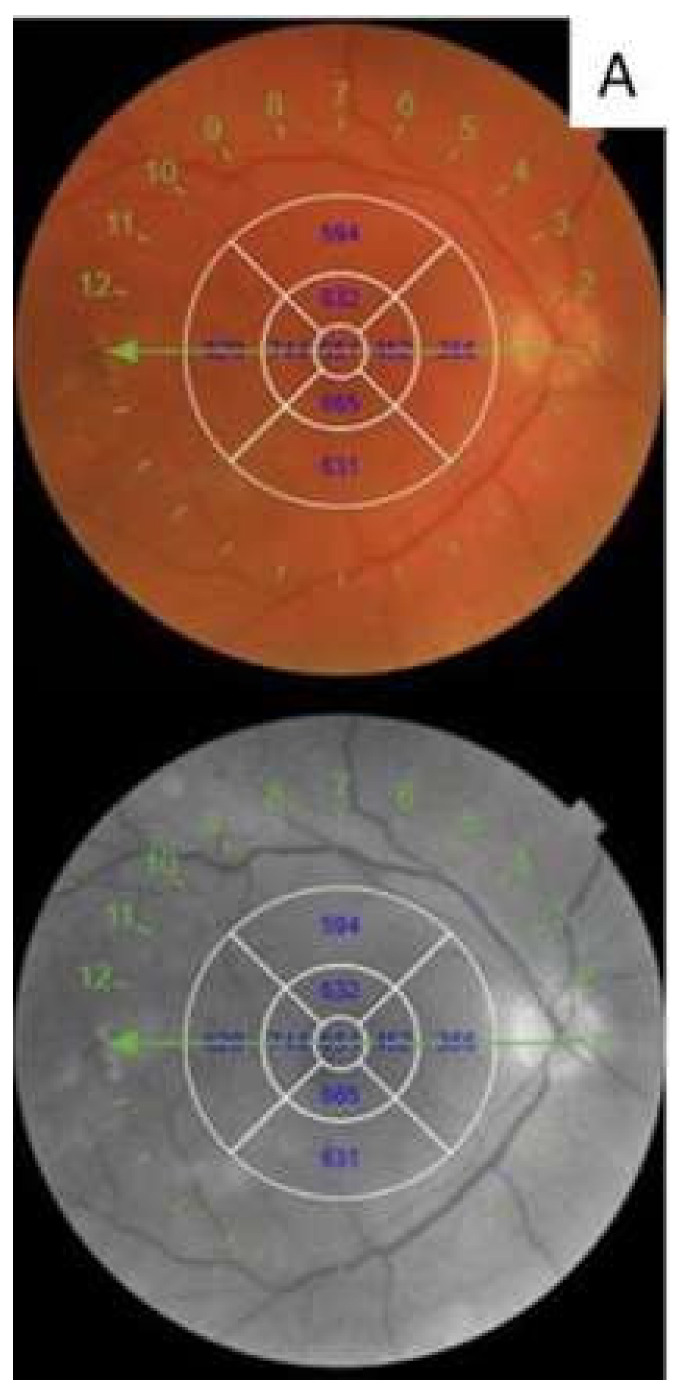
Retinal images. (**A**) Turnover of red dots observable at red free and color fundus photo. (**B**) White triangles indicates the hyper-reflective spots (HRS) in the retina layers of a patient with significant diabetic macular edema (DME), (**C**) intra-retinal cysts (IRC), (**D**) choroidal thickness, and (**E**) serous detachment of neuro-epithelium (SDN) height.

**Figure 2 diagnostics-10-00413-f002:**
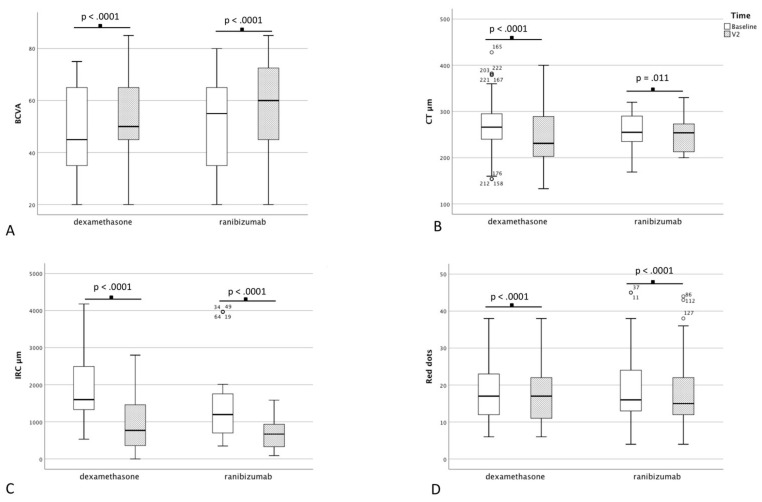
Box plot of: (**A**) best-corrected visual acuity, (**B**) choroidal thickness, (**C**) intra-retinal cysts, and (**D**) red dots.

**Figure 3 diagnostics-10-00413-f003:**
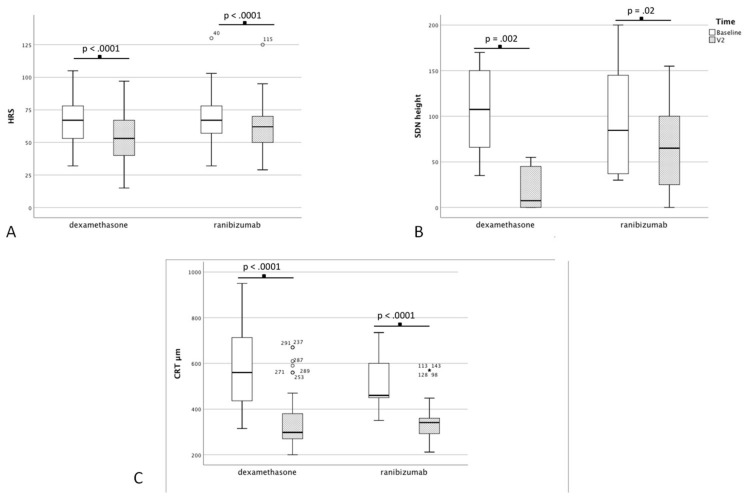
Box plot of: (**A**) hyper-reflective spots, (**B**) serous detachment of neuro-epithelium height, (**C**) central retinal thickness.

**Figure 4 diagnostics-10-00413-f004:**
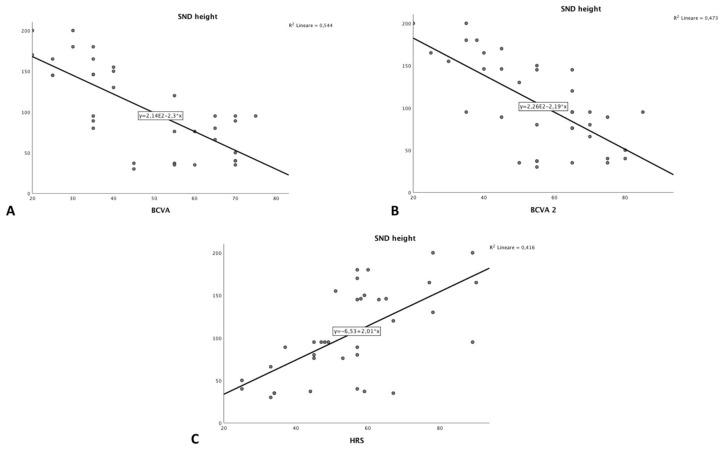
Scatter plot showing the relationship between SDN height and (**A**) BCVA baseline, (**B**) BCVA post-treatment and (**C**) HRS.

**Table 1 diagnostics-10-00413-t001:** Clinical characteristics of the study population.

Variables	Ranibizumab *n* = 75	Dexamethasone *n* = 81	*p*-Value
Age (years)	56.8 ± 7.6	59.9 ± 10.6	0.92 ^a^
Gender (male/female)	38/37	46/35	0.79 ^b^
Duration of diabetes (years)	13.9 ± 4.1	12.8 ± 4.3	0.85 ^a^
HbA1c (%)	8.47 ± 1.05	8.8 ± 1.06	0.81 ^a^
BCVA ETDRS	51.6 ± 17.1	47.8 ± 16.8	0.18 ^a^
CRT (µm)	511.1 ± 113.2	587.8 ± 177.9	0.20 ^a^
IRC (µm)	1467.2 ± 973.4	1719.2 ± 931.3	0.08 ^a^
CT	261.1 ± 33.1	271.5 ± 64.7	0.30 ^a^
SDN			
Number of patients (%)	18 (24.0%)	18 (22.2%)	0.78 ^b^
Height of SDN (µm)	100.5 ± 59.58	108.7 ± 47.4	0.38 ^a^
HRS	68.08 ± 19.8	66.41 ± 18.6	0.71 ^a^
Red Dots	19.24 ± 8.29	18.08 ± 7.6	0.45 ^a^

Legend: HbA1c—glycated hemoglobin; BCVA—best-corrected visual acuity; CRT—central retinal thickness; IRC—intra-retinal cysts; CT—choroidal thickness; SDN—serous neuro-retinal detachment; HRS—hyper-reflective spots. All data are reported as mean ± standard deviation. ^a^ Mann–Whitney test; ^b^ Chi Square test.

**Table 2 diagnostics-10-00413-t002:** Comparison between baseline and post-treatment (V2) data in the Ranibizumab group.

Parameters	Baseline	V2	*p*-Value ^a^
BCVA ETDRS	51.6 ± 17.1	56.9 ± 17.3	**<0.0001**
CRT (µm)	511.1 ± 113.2	342.4 ± 86.1	**<0.0001**
IRC (µm)	1467.2 ± 973.4	754.93 ± 447.2	**<0.0001**
CT	261.1 ± 33.1	259.2 ± 33.3	0.11
SDN height (µm)	100.5 ± 59.5	63.1 ± 46.8	**0.02**
HRS	68.1 ± 19.8	61.4 ± 17.9	**<0.0001**
Red Dots	19.24 ± 8.29	17.5 ± 8.5	**<0.0001**

Legend: BCVA—best-corrected visual acuity; CRT—central retinal thickness; IRC—intra-retinal cysts; CT—choroidal thickness; SDN—serous neuro-retinal detachment; HRS—hyper-reflective spots. V2: 1 month after completing the loading dose with three monthly injections. All data are reported as mean ± standard deviation. ^a^ Wilcoxon test. Bold characters for *p*-value < 0.05.

**Table 3 diagnostics-10-00413-t003:** Comparison between baseline and post-treatment (V2) data in the Dexamethasone group.

Parameters	Baseline	V2	*p*-Value ^a^
BCVA ETDRS	47.8 ± 16.8	55.4 ± 16.8	**<0.0001**
CRT (µm)	587.8 ± 177.9	338.9 ± 116.7	**<0.0001**
IRC (µm)	1719.2 ± 931.3	896.89 ± 715.6	**<0.0001**
CT	271.5 ± 64.7	242.9 ± 60.7	**<0.0001**
SDN height (µm)	108.7 ± 47.4	19.4 ± 23.5	**0.0002**
HRS	66.41 ± 18.6	53.1.9 ± 18.7	**<0.0001**
Red Dots	18.08 ± 7.6	17.2 ± 7.1	**<0.0001**

Legend: BCVA—best-corrected visual acuity; CRT—central retinal thickness; IRC—intra-retinal cysts; CT—choroidal thickness; SDN—serous neuro-retinal detachment; HRS—hyper-reflective spots.V2: 2 months after treatment. All data are reported as mean ± standard deviation. ^a^ Wilcoxon test. Bold characters for *p*-value < 0.05.

**Table 4 diagnostics-10-00413-t004:** Comparison of post-treatment data between the dexamethasone and ranibizumab groups.

Parameters	Ranibizumab	Dexamethasone	*p*-Value
BCVA	56.9 ± 17.3	55.4 ± 16.8	0.4 ^a^
CRT (µm)	342.4 ± 86.1	338.9 ± 116.7	0.06 ^a^
IRC (µm)	754.9 ± 447.2	896.9 ± 715.6	0.4 ^a^
CT (µm)	259.2 ± 33.3	242.9 ± 60.7	0.22 ^a^
SDN height (µm)	63.1 ± 46.8	19.4 ± 23.5	**0.03 ^a^**
SDN disappear, *n*	4 (22.2%)	16 (88.8%)	**0.0002 ^b^**
HRS	61.4 ± 17.9	53.1.9 ± 18.7	**0.01 ^a^**
Red Dots	17.5 ± 8.5	17.2 ± 7.1	0.78 ^a^

Legend: BCVA—best-corrected visual acuity; CRT—central retinal thickness; IRC—intra-retinal cysts; CT—choroidal thickness; SDN—serous neuro-retinal detachment; HRS—hyper-reflective spots. ^a^ Mann–Whitney test. ^b^ Chi-square test. Bold characters for *p*-value < 0.05.

**Table 5 diagnostics-10-00413-t005:** Multiple Regression analysis for baseline OCT biomarkers predicting factors associated with improvement of baseline visual acuity > 10 ETDRS letters at end of treatment.

Parameters	Coefficient	*p*-Value
CRT	−0.735	**0.0001**
IRC	−0.178	0.179
CT	−0.142	0.324
SND height	0.273	**0.032**
HRS	0.067	0.590

Legend: BCVA—best-corrected visual acuity; CRT—central retinal thickness; IRC—intra-retinal cysts; CT—choroidal thickness; SDN—serous neuro-retinal detachment; HRS—hyper-reflective spots.

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
