# Peer review of "The Application of Structural Retinal Biomarkers to Evaluate the Effect of Intravitreal Ranibizumab and Dexamethasone Intravitreal Implant on Treatment of Diabetic Macular Edema"

_diagnostics, 2020, doi:10.3390/diagnostics10060413_

Round 1
Reviewer 1 Report
The study compared the short-term outcomes of naive patients affected by diabetic macular oedema treated with monthly ranibizumab or dexamethasone implant.
Although interesting, the paper could benefit form a major revision.
First, the retrospective nature of the study is cited as limitation, but it has to be specified also in the introduction that the aim is to document only a short-term response to treatment. In this regard, an explanation of the follow-up interval chosen could be helpful for the readers.
Choroidal thickness is not a parameter of the classification mentioned, although is a recognized relevant marker. Please, correct and provide adequate reference fo choroidal thickness.
Some sentences in th paper need reference, e.g. lines 39-41 and 55.
Reference 6 should be changed with a more recent one.
Could you provide any indication of which are the paramenter that you considered to choose ranibizumab or dexamethasone implant?
Mentioning studies with longer follow-up could be helpful to discuss the effectiveness of both drugs
Author Response
Dear Reviewer 1,
We are grateful to you for your time and constructive comments on our manuscript.
We have amended the manuscript according your comments and valuable suggestions. Changes in the last version of the manuscript are reported as red tracked changes.
Below, we also provide a point-by-point response explaining how we have addressed each of your comments.
- The retrospective nature of the study is cited as limitation, but it has to be specified also in the introduction that the aim is to document only a short-term response to treatment.
Authors’ response:
Thanks for the comment and valuable suggestion. We defined the retrospective study design as the main limitation because it represents less statistical value than a prospective study. As suggested, we have specified in the Introduction section (from line 73 to line 75) that the aim of the study is to prove short-term outcomes variations in specific biomarkers in different treatments regimes.
- In this regard, an explanation of the follow-up interval chosen could be helpful for the readers.
Authors’ response:
We agree with your suggestion. We decided to evaluate the early response to anti-VEGF treatment and dexamethasone implant at the time of the maximum pharmacological effect of these drugs. Our aim was not to demonstrate the efficacy of these two drugs (by measuring outcomes such as improvement of BCVA) but to assess how they each induce changes in specific morphological biomarkers such as the presence of serous detachment of neuro-retina or hyper-reflective spots, given the recent literature on the role of inflammation in DME. In our opinion, these changes are best evaluated in the early months of the therapy. We added these aspects from line105 to line108.
- Choroidal thickness is not a parameter of the classification mentioned, although is a recognized relevant marker. Please, correct and provide adequate reference to choroidal thickness.
Authors’ response:
We have amended the text to reflect the latest classification of DME, and to include references to choroidal thickness. Please see the manuscript from line 58 to line 62, and from line 396 to line 419.
- Some sentences in the paper need reference, e.g. lines 39-41 and 55. Reference 6 should be changed with a more recent one.
Authors’ response:
As suggested, we have added references about the pathophysiology of DME and its treatment, and we have replaced reference 6 with a more recent one (now reference n.12) (line 48, line 68, from line 392 to line 393 and from line 437 to line 447).
- Could you provide any indication of which are the parameters that you considered to choose ranibizumab or dexamethasone implant?
Authors’ response:
The choice of the first treatment, in our clinical practice, is in accordance with the suggestions from the most recent guidelines (*). In particular, we prefer to treat with ranibizumab as the first choice in younger and phakic patients in the absence of cardiac contraindications. We usually prefer to treat with dexamethasone implant in pseudo-phakic patients, those with poor compliance to monthly treatment, and in patients with cardiac contraindications to ranibizumab or other anti-VEGF. We evaluated data from two homogeneous groups of patients for age, sex, and central macular thickens > 300 microns. These aspects have been clarified in the manuscript from line 88 to line 93.
Reference:
* Schmidt-Erfurth U, Garcia-Arumi J, Bandello F, et al. Guidelines for the Management of Diabetic Macular Edema by the European Society of Retina Specialists (EURETINA). Ophthalmologica. 2017;237(4):185‐222.
- Mentioning studies with longer follow-up could be helpful to discuss the effectiveness of both drugs.
Authors’ response:
Several studies have proven the effectiveness of both treatments in long term follow up. However, in our study we focused on the change of OCT biomarkers, in particular SDN and HRS, which are associated with intravitreal and retina hyperexpression of inflammatory cytokines. In this context there are only a few studies in the literature, all of which have relatively short follow up. (*) (from line 355 to line 363)
Reference:
*Nagaradh K, Gokarn P. Short term comparison results between two eyes of same individual treated with Dexamethasone implant and Ranibizumab in the management of naive Diabetic Macular Edema (DME). J Clin Ophthalmol 2020;4(1):215-221.
Looking forward to hearing from you.
Sincerely yours,
All coauthors

Reviewer 2 Report
In this manuscript, the authors evaluated the presence of specific swept-source OCT retinal biomarkers in patients with diabetic macular edema (DME) that guide treatment choice and predict therapeutic response. They analyzed treatment-naïve DME patients treated with 3 monthly intravitreal injections of ranibizumab or intravitreal implant of dexamethasone and showed the effectiveness of both treatment. They demonstrated DME associated with serous detachment of neuro-epithelium (SDN) and hyper-reflective spots (HRS) represents a specific inflammatory pattern 31 for which dexamethasone appears to be more effective. The manuscript is interesting, however I have some concerns that needs to be addressed.
- Line 124. Mean age was shown as 59 ±8.9 years. It is better to show to one decimal places. Also in throughout the manuscript.
- How did the authors select the treatment for each patient.
- Table 1. SDN (%) is not clear. Each group included 18 eyes with SDN. The description method like “SDN eyes (%), 18 eyes (22%)” are easy to understand.
- The authors mentioned that Table 4 showed a significant change in SDN and HRS between the two treatment groups after treatment. in two groups. However, it is difficult to understand what parameters they compared. Did they compare V2 SDN data 63.1 ± 46.8µm vs. 19.4 ± 23.5 µm or decreased level between baseline and V2?
- Table 4. Multiple regression model is useful for final analysis.
- Line 247. A complete resolution of SDN was noted in 93% of patients treated with dexamethasone after 3 months. The data are not shown in results. Were there any difference between two groups?
Author Response
Dear Reviewer 2,
We are grateful to you for your time and constructive comments on our manuscript.
We have implemented the manuscript following your comments and valuable suggestions. Changes in the last version of the manuscript are reported as red tracked changes.
Below, we also provide a point-by-point response explaining how we have addressed each of your comments.
Point 1: Line 124. Mean age was shown as 59 ±8.9 years. It is better to show to one decimal places. Also, in throughout the manuscript.
Authors’ replay:
Thanks for the suggestion. All data are now reported showing one decimal place in the tables and in the text.
Point 2: How did the authors select the treatment for each patient.
Authors’ replay:
This is a retrospective study, and we considered only patients with diabetic macular edema whom have received no other treatment previously. The choice of the first treatment, in our clinical practice, is in accordance with the suggestion from the most recent guidelines (*). In particular, we prefer to treat with ranibizumab as the first choice in younger and phakic patients in the absence of cardiac contraindications. We usually prefer to treat with dexamethasone implant pseudo-phakic patients, those with poor compliance to monthly treatment, and patients with cardiac contraindications to ranibizumab or other anti-VEGF. We evaluated data from two homogeneous groups of patients for age, sex, and central macular thickens > 300 microns. These aspects have been reported in the manuscript from line 88 lo line 93.
Reference: * Schmidt-Erfurth U, Garcia-Arumi J, Bandello F, et al. Guidelines for the Management of Diabetic Macular Edema by the European Society of Retina Specialists (EURETINA). Ophthalmologica. 2017;237(4):185‐222.
Point 3: Table 1. SDN (%) is not clear. Each group included 18 eyes with SDN. The description method like “SDN eyes (%), 18 eyes (22%)” are easy to understand.
Authors’ replay:
Thank you for your suggestion. We have now amended the table to include number of patients and percentage with SDN. We have also amended the data and the correct percentage is now reported (Table 1, line 159, line 193, line 198 and from line 202 to line 203).
Point 4: The authors mentioned that Table 4 showed a significant change in SDN and HRS between the two treatment groups after treatment. in two groups. However, it is difficult to understand what parameters they compared. Did they compare V2 SDN data 63.1 ± 46.8µm vs. 19.4 ± 23.5 µm or decreased level between baseline and V2?
Authors’ replay:
Thanks for the comment. In table 4 we compared data at the end of follow up in groups 1 and 2. We have now added this data into the table (Table 4 from line 205 to line 206).
Point 5: Table 4. Multiple regression model is useful for final analysis.
Authors’ replay:
We agree with your suggestion. We have performed a multiple regression analysis on the evaluated OCT biomarkers (table 5, from line 215 to line 226 and from line 31to line 33).
Point 6: Line 247. A complete resolution of SDN was noted in 93% of patients treated with dexamethasone after 3 months. The data are not shown in results. Were there any difference between two groups?
Authors’ replay:
We have added this data to the results, and included the number of patients with complete resolution. A significant difference was observed on the mean SDN height at the end of treatment. We have also amended the data and the correct percentage is now reported (from line 202 to line 203, table 4, line 316).
Looking forward to hearing from you.
Sincerely yours,
All coauthors

Reviewer 3 Report
In this study, Ceravolo et al. compared the efficacy of Ranibizumab vs. Dexamethasone implant (Ozurdex) in treatment of DME. There are several issues with this study.
1. The title is not descriptive of the study.
2. The introduction does not adequately deal with the motivation of the study.
3. The authors state that they evaluate OCT markers, but instead deal with comparative efficacy of Ranibizumab and Ozurdex. This aims of this study seem very unclear.
4. This was a retrospective, non-randomized, study. Many centers treat DME with anti-VEGF first, and consider Ozurdex in non-responders. Since no randomization has occurred, it is likely that the Ranibizumab group in this study has both responders and non-responders of anti-VEGF, whereas the Ozurdex group only has anti-VEGF non-responders. These circumstances give the comparisons of this study little, if any, value.
5. Table 4 states no data - only p-values. This is unfortunate.
6. Figure 5 has very poor image quality - I am unable to see the numbers stated on the figure, even after zooming in on the pdf.
7. The discussion is very confusing to read - there is no structure or flow.
Author Response
Dear Reviewer 3,
We are grateful to you for your time and constructive comments on our manuscript.
We have implemented the manuscript according your comments and valuable suggestions. Changes in the last version of the manuscript are reported as red tracked changes.
Below, we also provide a point-by-point response explaining how we have addressed each of your comments.
- The title is not descriptive of the study.
Authors’ replay:
We have modified the title to better reflect the study (from line 2 to line 5).
- The introduction does not adequately deal with the motivation of the study.
Authors’ replay:
Thanks for the comment. We have now modified the introduction focusing on the role and utility of OCT biomarkers, in particular SDN and HRS, and their correlation to the inflammatory mechanism in DME (from line 58 to line 62).
- The authors state that they evaluate OCT markers, but instead deal with comparative efficacy of Ranibizumab and Ozurdex. This aims of this study seem very unclear.
Authors’ replay:
We compared Lucentis and Ozurdex by examining the changes induced in specific OCT biomarkers at the end of treatment (CRT, IRC, SDN, HRS, CT). The aim was to evaluate the changes in these structural biomarkers and establish a possible clinical application in prediction of visual and morphological improvement with each treatment. Additionally, recent evidence has suggested a role for SDN and HRS as biomarkers of inflammation in DME. We documented a superiority of Ozurdex in reducing SDN. These aspects have been now better clarified from line 22 to line 24, from line 68 to line 75, and from line 106 to line 108.
- This was a retrospective, non-randomized, study. Many centers treat DME with anti-VEGF first, and consider Ozurdex in non-responders. Since no randomization has occurred, it is likely that the Ranibizumab group in this study has both responders and non-responders of anti-VEGF, whereas the Ozurdex group only has anti-VEGF non-responders. These circumstances give the comparisons of this study little, if any, value.
Authors’ replay:
In our study only “naïve” patients, whom have had no prior treatment, were enrolled. The choice of the first treatment in our clinical practice is in accordance with the suggestion from the most recent guidelines proposed by EURETINA (*). In particular, we prefer to treat with ranibizumab as the first choice in younger and phakic patients without cardiac contraindications. We usually prefer to treat with dexamethasone implant in pseudo-phakic patients, those with poor compliance to monthly treatment, and patients with cardiac contraindications to ranibizumab or other anti-VEGF. For this reason, because the patients are naïve, those in the group treated with Ozurdex are not necessarily anti-VEGF non-responders, but they are patients with a contraindication to anti-VEGF as a first line therapy. These aspects have been better clarified from line 88 to line 93.
Reference
* Schmidt-Erfurth U, Garcia-Arumi J, Bandello F, et al. Guidelines for the Management of Diabetic Macular Edema by the European Society of Retina Specialists (EURETINA). Ophthalmologica. 2017;237(4):185‐222.
- Table 4 states no data – only p-values. This is unfortunate.
Authors’ replay:
We agree with the comment. We have now changed the table 4, reporting all data (from line 205 to line 206).
- Figure 5 has very poor image quality - I am unable to see the numbers stated on the figure, even after zooming in on the pdf.
Authors’ replay:
We agree with the comment. We have replaced the image with a higher quality one (from line 117 to line 122).
- The discussion is very confusing to read - there is no structure or flow.
Authors’ replay:
Thanks for the comment. As suggested, we have modified the discussion, focusing on the role of every OCT biomarker, and discussing the results and their application. Additionally, the structure and the flow have been improved (from line 235 to line 363).
Looking forward to hearing from you.
Sincerely yours,
All coauthors

Round 2
Reviewer 1 Report
The manuscript improved after the first revision. However, it could further improve with additional changes.
- lines 47-49: the two sentences could be merged as the text appears repetitive
- line 50: add "subretinal"
- line 64-65: I would change "categorize and grade" with "characterize"
- line 73: I would change "ocular tone" with "intraocular pressure"
- line 90: specify when cataract is exclusion criteria (I do not think all patients with cataract were excluded)
- line 95: it seems that all pseudophaic patients received dexamethasone implant. Please, clarify
- lines 105-108: there is no explanation as written in the Authors' response
- Intraretinal cysts measurment: in the paper cited only the largest intraretinal cyst is measured. Why do the Authors choose to measure all the ICR within 3000 microns from the foveal centre? Please clarify and add reference
- lines 130-131: rephrase
- 137: please change "fovea" to "foveal centre"
- a small introduction in the initial part of discussion could be beneficial
- lines 262-264, 265-267, 318-319, 335: please, add reference
- lines 268, 270, 277-279, 332: please, rephrase
- lines 323-324, 333-334: I would delete as it is repetitive
- line 325: the desription of HRS' features would be better in the method session
- lines 339-343: I would delete this sentence as it does not add anything to the discussion
- lines 361-362 "speculating that short-term effects can predict the therapeutic 361 outcome": this sentence need to be discussed and motivated as it is not supported by the results of the study. Otherwise, in absence of supporting evidence, it should be deleted
Author Response
Dear Reviewer,
We are grateful to you for your time and constructive comments on our manuscript.
We have amended the manuscript according your comments. Changes in the last version of the manuscript are reported as red tracked changes.
Below, we also provide a point-by-point response explaining how we have addressed each of your comments.
1. lines 47-49: the two sentences could be merged as the text appears repetitive.
Authors’ replay:
As suggested, we have amended the text (line 43).
2. line 50: add "subretinal"
Authors’ replay:
As suggested, we have amended the text (line 44).
3. line 64-65: I would change “categorize and grade” with “characterize”.
Authors’ replay:
As suggested, we have amended the text (line 55).
4. line 73: I would change "ocular tone" with "intraocular pressure".
Authors’ replay:
As suggested, we have amended the text (line 63).
5. line 90: specify when cataract is exclusion criteria (I do not think all patients with cataract were excluded).
Authors’ replay:
We excluded patients with cataracts that affected visual acuity, to evaluate only patients with a reduction of visual acuity due to diabetic macular edema (line 80).
6. line 95: it seems that all pseudophakic patients received dexamethasone implant. Please, clarify.
Authors’ replay:
As suggested, we have clarified it (from line 85 to line 86).
7. lines 105-108: there is no explanation as written in the Authors' response.
Authors’ replay:
As suggested, we have clarified it (from line 101 to line 102).
8. Intraretinal cysts measurment: in the paper cited only the largest intraretinal cyst is measured. Why do the Authors choose to measure all the ICR within 3000 microns from the foveal centre? Please clarify and add reference.
Authors’ replay:
The new classification of DME proposed by the ESASO group is useful in clinical practice, but in our opinion, in a scientific study, to assess the volume of the intraretinal fluid, a specific measurement of a single cyst is more accurate. We chose to evaluate edema in an area of 3000 microns to include the central and paracentral location of the cysts, as well as to evaluate HRS (reference n° 15).
9. lines 130-131: rephrase.
Authors’ replay:
We have amended the text (from line 127 to line 129).
- 137: please change "fovea" to "foveal centre".
Authors’ replay:
As suggested, we have changed it (line 141).
- a small introduction in the initial part of discussion could be beneficial.
Authors’ replay:
As suggested, we have implemented the discussion (from line 247 to line 249).
- lines 262-264, 265-267, 318-319, 335: please, add reference.
Authors’ replay:
As suggested, we have added references (from line 239 to line 240, from line 240 to line 243, line 292-293, line 311).
- lines 268, 270, 277-279, 332: please, rephrase.
Authors’ replay:
As suggested, we have rephrased it (from line 247 to line 249, from line 250 to line 252, from line 257 to line 259, from line 307 to line 311).
- lines 323-324, 333-334: I would delete as it is repetitive.
Authors’ replay:
As suggested, we have deleted it (from line 296 to line 299; from line 306 to line 308).
- line 325: the description of HRS' features would be better in the method session.
Authors’ replay:
As suggested, we have added HRS description in the methods (before line 298-299, now from line 119 to line 121).
- lines 339-343: I would delete this sentence as it does not add anything to the discussion.
Authors’ replay:
As suggested, we have deleted the sentence (from line 313 to line 317).
- lines 361-362 "speculating that short-term effects can predict the therapeutic 361 outcome": this sentence need to be discussed and motivated as it is not supported by the results of the study. Otherwise, in absence of supporting evidence, it should be deleted.
Authors’ replay:
As suggested, we have deleted the sentence (from line 328 to line 335).
Looking forward to hearing from you,
All coauthors.

Reviewer 2 Report
All questions were replyed at revised version.
Author Response
Dear Reviewer,
We are grateful to you for your help and valuable comments. The manuscript was revised by one of the co-authors that is a native English speaker.
Looking forward to hearing from you
Best wishes,
All coauthors
Reviewer 3 Report
Thank you for your revision. The manuscript has improved. I have no further comments.
Author Response

(The authors gave the same response as above.)

Round 3
Reviewer 1 Report
The paper significanlty improved after the revisions.